# A Three-Parameter Weibull Distribution Method to Determine the Fracture Property of PMMA Bone Cement

**DOI:** 10.3390/polym14173589

**Published:** 2022-08-30

**Authors:** Lielie Li, Hekai Cao, Junfeng Guan, Shuanghua He, Lihua Niu, Huaizhong Liu

**Affiliations:** 1School of Civil Engineering and Communication, North China University of Water Resources and Electric Power, Zhengzhou 450045, China; 2State Key Laboratory of Hydraulics and Mountain River Engineering, Sichuan University, Chengdu 610065, China

**Keywords:** PMMA bone cement, Weibull distribution, tensile strength, fracture toughness

## Abstract

Poly (methyl methacrylate) (PMMA) bone cement is an excellent biological material for anchoring joint replacements. Tensile strength $f_{t}$ and fracture toughness $K_{\mathrm{IC}}$ have a considerable impact on its application and service life. Considering the variability of PMMA bone cement, a three-parameter Weibull distribution method is suggested in the current study to evaluate its tensile strength and fracture toughness distribution. The coefficients of variation for tensile strength and fracture toughness were the minimum when the characteristic crack of PMMA bone cement was $\alpha_{\mathrm{ch}}^{*}=8d_{\mathrm{av}}$. Using the simple equation $\alpha_{\mathrm{ch}}^{*}=8d_{\mathrm{av}}$ and fictitious crack length $\Delta \alpha_{\mathrm{fic}}=1.0d_{\mathrm{av}}$, the mean value μ (= 43.23 MPa), minimum value $f_{t}^{\min}$ (= 26.29 MPa), standard deviation σ (= 6.42 MPa) of tensile strength, and these values of fracture toughness (μ = 1.77 $\mathrm{MPa}\cdot m^{1/2}$, $K_{\mathrm{IC}}^{\min}$ = 1.02 $\mathrm{MPa}\cdot m^{1/2}$, σ = 0.2644 $\mathrm{MPa}\cdot m^{1/2}$) were determined simultaneously through experimental data from a wedge splitting test. Based on the statistical analysis, the prediction line between peak load $P_{\max}$ and equivalent area $A_{e}^{1}\left(A_{e}^{2}\right)$ was obtained with 95% reliability. Nearly all experimental data are located within the scope of a 95% confidence interval. Furthermore, relationships were established between tensile strength, fracture toughness, and peak load $P_{\max}$. Consequently, it was revealed that peak load might be used to easily obtain PMMA bone cement fracture characteristics. Finally, the critical geometric dimension value of the PMMA bone cement sample with a linear elastic fracture was estimated.

## 1. Introduction

Poly (methyl methacrylate) (PMMA) bone cement is a typical quasi-brittle material commonly used in total joint replacements (TJRs) [1,2,3,4,5,6,7]. It can be anchored to continuous bone for applications such as fixation of the implant, delivery of antibiotics, filling of bone defects, etc. [8,9]. Fracture toughness $K_{\mathrm{IC}}$ and tensile strength $f_{t}$ of PMMA bone cement are two essential indicators for determining stability and service life performance. The relevant information and mechanical characteristics of PMMA bone cement in the existing literature are presented in Table 1. As shown in Table 1, the test results of $K_{\mathrm{IC}}$ and $f_{t}$ of PMMA bone cement have significant discreteness. Therefore, it is urgent and important to define a reasonable distribution function for statistical analysis to help obtain the exact parameter properties of PMMA bone cement.

The Weibull distribution function, as a skewed function, is commonly used to investigate the discreteness of the parameters of quasi-brittle materials. For example, Amaral investigated the flexural strength of granite through using the Weibull distribution function. Based on statistical results, engineers can specify specific types of brittle building materials to improve the safety of construction projects [22]. Blasi demonstrated, using Weibull statistics, that the size effect has a significant impact on the flexural strength of marble [23]. Li utilized the Weibull function to calculate the fracture toughness and tensile strength of concrete and predicted peak loads with 95% reliability [24]. Xu investigated the size effect of concrete by using the Weibull function. The better the uniformity of concrete is, the more obvious the size effect will be [25]. Lei reformulated and validated the three-parameter Weibull statistical fracture theory for uniaxial flexure of prismatic beams by analyzing the strength and experimental data of different ceramics [26]. Hu used the Weibull distribution to investigate the microcrack toughness of Yttria-stabilized zirconia (3Y-TZP) [27]. Gorjan indicated that the Weibull distribution provides the best accuracy for strength scattering with high alumina ceramics, outperforming the normal, log-normal, and Gamma distributions [28]. The Weibull function has also been used to investigate the fatigue properties of concrete [29,30]. According to Bala’s research, the fatigue life of a composite asphalt mixture follows the Weibull distribution [31]. Jin investigated the compressive strength variability of tungsten particle (Wp)-reinforced Zr-based bulk metallic glass composites using the Weibull distribution [32]. Furthermore, the Weibull function was employed to analyze polymer properties. In this context, Carmona investigated the failure probability of natural *Luffa cylindrica* fibers using the Weibull distribution function [33], whereas Sia utilized the Weibull distribution to quantify variations in the tensile strength of pineapple leaf fibers [34]. Wang investigated the effect of the length and diameter of bamboo fibers on tensile strength by using a modified Weibull model. It was shown that the accuracy of the Weibull model in terms of strength and predicted size correlation of bamboo fibers was satisfactory [35]. Wang described the statistical distribution of the critical energy release rate (Gc) of the transverse layer (the transverse layer) by using a two-parameter Weibull function. Combined with the numerical model, the mechanical behavior of the laminated composite formed by unidirectional fiber reinforced laminae can be calculated [36]. Equivalent fracture toughness of EPOLAM 2025 CT epoxy resin was also analyzed using the Weibull distribution [37].

Previous studies have clearly revealed that the parameter characteristics of quasi-brittle materials and high polymers can be statistically analyzed with the Weibull distribution function. However, PMMA bone cement is a high polymer quasi-brittle material with uncertain distribution characteristics regarding its fracture toughness and tensile strength parameters. In this paper, based on the fracture test results for wedge-splitting specimens (WS) [38], $K_{\mathrm{IC}}$ and $f_{t}$ of PMMA bone cement were investigated using the three-parameter Weibull distribution function; the mean and variance of its fracture toughness and tensile strength parameters were obtained. According to the findings, the minimum coefficient of variation (CV) was obtained when the relative characteristic crack was $\alpha_{\mathrm{ch}}^{*}=8d_{\mathrm{av}}$. Using the constant value of β (= 1.0), the quasi-brittle fracture of PMMA bone cement was predicted. Furthermore, the peak loads $P_{\max}$ with a specified 95% reliability were predicted by using three-parameter Weibull distribution analysis.

## 2. The Theoretical Background of PMMA Distribution Characteristics

### 2.1. Intrinsic Causes of PMMA Fracture Discreteness

A previous experimental study on the subject revealed that, since the same batch of PMMA bone cement specimens with the same size and loading procedure were used on the same equipment, the recorded peak load $P_{\max}$ maintained its discreteness [38]. As illustrated in Figure 1, the fictitious crack growth in PMMA bone cement specimens of the same size was discrete due to the random distribution of material particles, which caused the unpredictability of $P_{\max}$. This meant that separation was the key feature of PMMA bone cement. The variation dispersion coefficient β is introduced in the current paper to investigate this discreteness, and Equation (1) was utilized to establish the relationship between the fictitious crack growth $\Delta a_{\mathrm{fic}}$ and the average particle size $d_{\mathrm{av}}$ at the peak load $P_{\max}$ [39]. As shown in Figure 2, the three-parameter Weibull distribution function was used to characterize the distribution characteristics of the variation dispersion coefficient β to assess the statistical characteristics of the fracture strength of PMMA bone cement. Further, $\Delta a_{\mathrm{fic}}$ and $P_{\max}$ related to β match the three-parameter Weibull distribution.
(1)$$\Delta \alpha_{\mathrm{fic}}=\beta d_{\mathrm{av}}$$

The Weibull distribution function is commonly used to characterize the statistical distribution of parameters of materials such as rocks, concrete, ceramics, and polymers. This section provides a brief description of the three-parameter Weibull distribution function. The parameters x ($x_{1}$, $x_{2}$, $x_{3}$...$x_{i}$...$x_{N}$) were adjusted to fit the three-parameter Weibull distribution function. The probability density of parameter x is calculated by Equation (2) as follows:(2)$$P_{f}(x\leq x_{i})=F_{i}=\frac{N_{i}}{N}$$
where $P_{f}(x\leq x_{i})$ is equal to or less than the cumulative frequency of x, N represents the total number of parameters, and $N_{i}$ is equal to or less than the value of $x_{i}$.

The probability density function of the three-parameter Weibull distribution can be expressed as follows:(3)$$P_{f}(x)=f(x)=\frac{K}{\omega }{\left(\frac{x-x_{\min}}{\omega }\right)}^{K-1}\exp\left[-{\left(\frac{x-x_{\min}}{\omega }\right)}^{K}\right]{}^{}x\geq x_{\min}$$

The corresponding probability distribution function is shown in Equation (4)
(4)$$P_{f}(x\leq x_{i})=F(x\leq x_{i})=1.0-\exp\left[-{\left(\frac{x-x_{\min}}{\omega }\right)}^{K}\right]{}^{}x\geq x_{\min}$$
where ω is the size parameter of the Weibull distribution, K is the shape parameter of the Weibull distribution, and x is the position parameter (it is the minimum value obtained by fitting the parameter x, no parameter is smaller than this point).

The mean value μ and standard deviation σσ of the three-parameter Weibull distribution function is obtained by Equations (5) and (6):(5)$$\mu =E(x)=x_{\min}+\omega \Gamma \left(1+\frac{1}{K}\right)$$
(6)$$\sigma^{2}=\sqrt{\omega^{2}\left[\Gamma \left(1+\frac{1}{K}\right)\right]-\Gamma^{2}\left(1+\frac{1}{K}\right)}$$
where Γ is the gamma function.

### 2.2. A Brief Description of the Boundary Effect Model (BEM)

Wedge-splitting specimens (WS) [11,13], three-point bending specimens [10], and chevron-notched short-rod specimens [14,15,16] are commonly used in laboratory testing to examine the fracture parameters of PMMA bone cement. The tensile strength of PMMA bone cement was obtained through four-point bending specimens [19] and dumb-bell specimens [18,21]. A large number of experiments are required to obtain the necessary parameters, which increases costs and the time required for the experiments. It is proposed in this section that the BEM model be adapted to PMMA bone cement specimens; thus, the two parameters of fracture toughness and tensile strength can be calculated simultaneously only based on WS test results. In this paper, the distribution form of the nominal stress at the crack tip was assumed to be rectangular [39], as shown in Figure 3. Furthermore, the BEM considering the variation dispersion coefficient β, the average aggregate size $d_{\mathrm{av}}$, and the peak load $P_{\max}$ is given in Equation (7) [39,40,41,42].
(7)$$\frac{1}{\sigma_{n}^{2}\left(P_{\max},\Delta \alpha_{\mathrm{fic}}\right)}=\frac{1}{\sigma_{n}^{2}\left(P_{\max},\beta d_{\mathrm{av}}\right)}=\frac{1}{f_{t}^{2}}+\frac{4a_{e}}{K_{\mathrm{IC}}^{2}}$$
where $\sigma_{n}$ represents nominal stress.

For the WS specimens, the equivalent crack $a_{e}$ can be determined by the following Equation [39,40,41,42]:(8)$$a_{e}={\left[\frac{\frac{2{\left(1-\alpha \right)}^{2}}{2+\alpha }\times Y\left(\alpha \right)}{1.12}\right]}^{2}\times a_{0}$$
where α is the ratio of $a_{0}$ to *W*, $\alpha =a_{0}/W$, $a_{0}$ is the initial crack length, and $Y\left(\alpha \right)$ is the geometric factor. The expression of $Y\left(\alpha \right)$ is as follows [39,40,41,42]:(9)$$Y\left(\alpha \right)=\frac{\left(2+\alpha \right)\times \left(0.866+4.64\alpha -13.32\alpha^{2}+14.72\alpha^{3}-5.6\alpha^{4}\right)}{4\sqrt{\pi \alpha }{\left(1-\alpha \right)}^{3/2}}$$

According to the stress distribution form in Figure 4, the expression for nominal stress $\sigma_{n}$ can be derived by balancing the force and moment of force according to the following Equation [39,40,41,42]:(10)$$\sigma_{n}\left(P_{\max},\beta d_{\mathrm{av}}\right)=\frac{P_{\max}\left(3W_{2}+W_{1}\right)}{6B\left(\frac{W_{1}^{2}}{6}+\frac{W_{1}}{6}\left(\beta d_{\mathrm{av}}\right)+\left(\frac{W-a_{0}}{2}\right)\left(\beta d_{\mathrm{av}}\right)\right)}$$
where *B* is the width of the specimen and *W* is the height of the specimen.
(11)$$W_{1}=W-a_{0}-\beta d_{\mathrm{av}}$$
(12)$$W_{2}=W+a_{0}+\beta d_{\mathrm{av}}$$

In the BEM, the ratio of characteristic crack length $\alpha_{\mathrm{ch}}^{*}$ to average aggregate size $d_{\mathrm{av}}$ is a constant *C*, obtained as follows [39]:(13)$$C=\alpha_{\mathrm{ch}}^{*}/d_{\mathrm{av}}=0.25{\left(K_{\mathrm{IC}}/f_{t}\right)}^{2}/d_{\mathrm{av}}$$

Combining Equations (7), (13), and (14), (15) provides the following:(14)$$f_{t}=\sigma_{n}\left(P_{\max},\beta d_{\mathrm{av}}\right)\sqrt{1+\frac{a_{e}}{Cd_{\mathrm{av}}}}$$
(15)$$K_{\mathrm{IC}}=2\sigma_{n}\left(P_{\max},\beta d_{\mathrm{av}}\right)\sqrt{a_{e}+Cd_{\mathrm{av}}}$$

Substituting Equation (10) into Equations (14)–(17) obtains:(16)$$f_{t}=\frac{P_{\max}\left(3W_{2}+W_{1}\right)}{6B\left(\frac{W_{1}^{2}}{6}+\frac{W_{1}}{6}\left(\beta d_{\mathrm{av}}\right)+\left(\frac{W-a_{0}}{2}\right)\left(\beta d_{\mathrm{av}}\right)\right)}\times \sqrt{1+\frac{a_{e}}{Cd_{\mathrm{av}}}}$$
(17)$$K_{\mathrm{IC}}=\frac{P_{\max}\left(3W_{2}+W_{1}\right)}{3B\left(\frac{W_{1}^{2}}{6}+\frac{W_{1}}{6}\left(\beta d_{\mathrm{av}}\right)+\left(\frac{W-a_{0}}{2}\right)\left(\beta d_{\mathrm{av}}\right)\right)}\times \sqrt{a_{e}+Cd_{\mathrm{av}}}$$

Individual fracture toughness and tensile strength can be calculated according to *C*, β, $d_{\mathrm{av}}$, $P_{\max}$_,_ and the geometric parameters of the specimens, as seen from Equations (16) and (17). The dispersion coefficient β, fictitious crack growth $\Delta a_{\mathrm{fic}}$_,_ and peak load $P_{\max}$ all provide the three-parameter Weibull function distributions. Therefore, tensile strength $f_{t}$ and fracture toughness $K_{\mathrm{IC}}$ also provide the three-parameter Weibull function distributions. Additionally, the mean value μ and standard deviation σ of the statistical distribution for PMMA bone cement strength fracture parameters were obtained according to Equations (5) and (6).

## 3. Statistical Analysis of Test Data

### 3.1. Raw Data of the Experiment

A specific quantity of the experimental sample data set is needed to test the validity of the statistical analysis results. In this paper, experimental data from PMMA bone cement wedge separation samples by Merta were selected for analysis as a data set [38]. The geometric information of the specimens and the test results of the load peak value $P_{\max}$ are presented in Table 2. The width *B* of the test specimens was 6 mm. The α (the ratio of $a_{0}$ to *W*) variation range of the specimen height *W* (= 15 mm) was 0.13~0.53. The α (the ratio of $a_{0}$ to *W*) variation range of the specimen height *W* (= 22 mm) was 0.09~0.64. The α (the ratio of $a_{0}$ to *W*) variation range of the specimen height *W* (= 29 mm) was 0.07~0.69. The α (the ratio of $a_{0}$ to *W*) variation range of the specimen height *W* (= 36 mm) was 0.05~0.81. For each combination of $a_{0}$ and *W*, 5 to 9 specimens were produced, with a total of 160 specimens, and all specimens with cracks larger than 2 mm were removed, thus leaving 96 specimens. The numbering rule for specimens in this paper is E + “height” + “-initial crack length” for the convenience of expression. As an example, E15-2 represents a specimen with a height of 15 mm and a 2 mm initial crack length. In the analysis, the constant *C* was taken as 0.5, 1, 1.5, 2, 2.5, 3, 4, 5, 6, 8, 9, and 10, respectively; β adopted the constant method to take a value (= 1, 2, 3, 4). According to reports in the relevant literature, the aggregate size *d* of PMMA bone cement ranges from 5 to 100 μm40, with average aggregate size $d_{\mathrm{av}}$ equal to 52.5 μm. Fracture toughness $K_{\mathrm{IC}}$ and tensile strength $f_{t}$ were calculated according to Equations (16) and (17). The statistical distribution results of fracture toughness $K_{\mathrm{IC}}$ and tensile strength $f_{t}$ were then obtained using the three-parameter Weibull distribution function.

### 3.2. Raw Data of the Experiment

Based on the findings of the peak load $P_{\max}$ test on 96 PMMA bone cement WS specimens and using Equations (16) and (17), 96 values of tensile strength $f_{t}$ and fracture toughness $K_{\mathrm{IC}}$ were calculated for each combination of *C*, β, and $d_{\mathrm{av}}$ values, respectively. Further analyses were performed using the three-parameter Weibull function, and Equations (5) and (6) were used to obtain the statistical mean value μ, standard deviation σ, and coefficient of variation CV (= σ/μ) for fracture strength parameters of PMMA bone cement. The three-parameter Weibull distributions of tensile strength $f_{t}$ and fracture toughness $K_{\mathrm{IC}}$ for *C* = 8.0 at $d_{\mathrm{av}}=52.5\;\mu m$ are shown in Figure 4 and Figure 5, which are constrained by article space. It can be clearly seen that the distribution curve is in good agreement with the experimental results.

The statistical findings for the experimental data obtained from Merta using the three-parameter Weibull distribution are listed in Table 3 [38]. The coefficient of variation CV curves of tensile strength $f_{t}$ and fracture toughness $K_{\mathrm{IC}}$, with the respective *C* value, are shown in Figure 6. As can be observed, the coefficient of variation CV of $f_{t}$ reaches a lesser or minimal value when *C* = 8.0, regardless of the value of β. As shown in Table 3, the value β has little impact on the statistical characteristics of fracture strength when the value of *C* is constant. For the convenience of calculation, the constant *C* for the PMMA bone cement specimen was 8.0, and the variation dispersion coefficient β was 1.0. The calculated average tensile strength $f_{t}$ of PMMA bone cement was 43.23MPa. It is located in the range of 33.0 MPa~ 51.4 MPa for tensile strength $f_{t}$ which has been reported in related literature [17,18,19,20,21]. The average value of fracture toughness $K_{\mathrm{IC}}$ was 1.77 $\mathrm{MPa}\cdot m^{1/2}$, which likewise falls within the range of 1.0 $\mathrm{MPa}\cdot m^{1/2}$ ~ 2.7 $\mathrm{MPa}\cdot m^{1/2}$ as reported by related literature [10,11,12,13,14,15,16,17]. In addition, the minimum value $f_{t}^{\min}$ of tensile strength was 26.29 MPa. The minimum value $K_{\mathrm{IC}}^{\min}$ of fracture toughness was 1.02 $\mathrm{MPa}\cdot m^{1/2}$. Minimum values of tensile strength and fracture toughness provided a basis for evaluating the safety of PMMA bone cement.

## 4. Three-Parameter Weibull Distribution Prediction of PMMA Bone Cement Fracture

The constant *C* was taken as 8.0 and the variation dispersion coefficient β as 1.0 based on the analysis results in Section 3. The statistical distribution results of tensile strength $f_{t}$ and fracture toughness $K_{\mathrm{IC}}$ were obtained from the three-parameter Weibull distribution function. Combined with Equation (7), the complete fracture prediction curve of PMMA bone cement can be predicted.

The lower limit of the material parameter can be established based on the minimum value of the parameter, according to the three-parameter Weibull distribution principle. Namely, according to the minimum value of tensile strength $f_{t}$ and fracture toughness $K_{\mathrm{IC}}$, a minimum safety line is determined as the minimum safety control index. According to the complete fracture prediction line of PMMA bone cement, the tensile strength ($f_{t}$) control zone ($a_{e}/a_{\mathrm{ch}}^{*}\leq 0.1$), the quasi-brittle fracture control zone ($0.1\leq a_{e}/a_{\mathrm{ch}}^{*}\leq 10$), and the fracture toughness ($K_{\mathrm{IC}}$) control zone ($a_{e}/a_{\mathrm{ch}}^{*}\geq 10$) can be obtained.

Figure 7 depicts the predicted results of the 95% confidence interval (μ±2σ) of the PMMA bone cement WS specimen based on the mean value and standard deviation. By analyzing the minimum value, the lower limit of the material parameter was determined. As shown in Figure 7, the PMMA bone cement WS data points in the laboratory experiments are all in the quasi-brittle fracture control zone and above the lower limit fracture failure curve fitted by the minimum value. Almost all experimental data are within the 95% confidence interval.

By transforming Equations (16) and (17), a simplified prediction model for the tensile strength $f_{t}$ and fracture toughness $K_{\mathrm{IC}}$ of PMMA bone cement WS specimens can be obtained. Tensile strength $f_{t}=P_{\max}/A_{e}^{1}$ and fracture toughness $K_{\mathrm{IC}}=P_{\max}/A_{e}^{2}$, with respect to the equivalent area $A_{e}^{1}\left(A_{e}^{2}\right)$ and the peak load $P_{\max}$, can be obtained as shown in Equations (18) and (19). Conversely, the peak load $P_{\max}$ of the test piece can also be calculated from the equivalent area $A_{e}^{1}\left(A_{e}^{2}\right)$ and $f_{t}$ ($K_{\mathrm{IC}}$). Mean value μ and standard deviation σ of fracture strength parameters were obtained according to three-parameter Weibull distribution statistics. As shown in Figure 8, the predicted results of the peak load $P_{\max}$ for the PMMA bone cement WS specimen with a 95% confidence interval (μ±2σ) were obtained at the lower limit of the prediction of peak load $P_{\max}$. As can be seen in Figure 8, the vast majority of data are within the 95% confidence range, and all experimental data were above the minimum fit’s lower limit prediction curve.
(18)$$P_{\max}=f_{t}\frac{6B\left(\frac{W_{1}^{2}}{6}+\frac{W_{1}}{6}\left(\beta d_{\mathrm{av}}\right)+\left(\frac{W-a_{0}}{2}\right)\left(\beta d_{\mathrm{av}}\right)\right)}{\left(3W_{2}+W_{1}\right)\times \sqrt{1+\frac{a_{e}}{Cd_{\mathrm{av}}}}}=f_{t}A_{e}^{1}$$
(19)$$P_{\max}=K_{\mathrm{IC}}\frac{6B\left(\frac{W_{1}^{2}}{6}+\frac{W_{1}}{6}\left(\beta d_{\mathrm{av}}\right)+\left(\frac{W-a_{0}}{2}\right)\left(\beta d_{\mathrm{av}}\right)\right)}{1.12\times \sqrt{\pi }\left(3W_{2}+W_{1}\right)\times \sqrt{a_{e}+Cd_{\mathrm{av}}}}=K_{\mathrm{IC}}A_{e}^{2}$$

## 5. Discussion

The situation of β= 1.0, 2.0, 3.0, and 4.0 is discussed in this section according to Table 3. When C=8, the mean value, minimum value, standard deviation σ, coefficient of variation CV of tensile strength $f_{t}$, and these values of fracture toughness $K_{\mathrm{IC}}$ presented slight variations with increases in β. For example, when β was increased from 1 to 4, the mean values of $f_{t}$ only varied from 42.43 MPa to 43.23 MPa, and values of fracture toughness $K_{\mathrm{IC}}$ only varied from 1.74 $\mathrm{MPa}\cdot m^{1/2}$ to 1.77 $\mathrm{MPa}\cdot m^{1/2}$. Limited by space, only the destruction full curve and prediction curves for β=4 were plotted, as shown in Figure 9 and Figure 10. Nearly all experimental data are also located within the scope of a 95% confidence interval. However, as shown in Figure 4, when β=1 the correlation coefficients of $f_{t}$ and $K_{\mathrm{IC}}$ both have the maximum value of 0.9867 and 0.9191, respectively. With the increase in β, the correlation coefficients of $f_{t}$ and $K_{\mathrm{IC}}$ present a downtrend. Thus, it is reasonable to take β=1 for the convenience of calculation and application. In addition, more test data are required to demonstrate the correctness of three-parameter Weibull distribution of material parameters in the future.

## 6. Conclusions

A three-parameter Weibull distribution approach was used to assess the fracture property of PMMA bone cement based on the experimental results of wedge-splitting specimens of PMMA bone cement. During the process, the characteristic crack of PMMA bone cement ($\alpha_{\mathrm{ch}}^{*}=Cd_{\mathrm{av}}$
$=0.5d_{\mathrm{av}}$, $1.0d_{\mathrm{av}}$, $1.5d_{\mathrm{av}}$, $2.0d_{\mathrm{av}}$, $2.5d_{\mathrm{av}}$, $3.0d_{\mathrm{av}}$, $4.0d_{\mathrm{av}}$, $5.0d_{\mathrm{av}}$, $6.0d_{\mathrm{av}}$, $7.0d_{\mathrm{av}}$, $8.0d_{\mathrm{av}}$, $9.0d_{\mathrm{av}}$, $10.0d_{\mathrm{av}}$) at peak loads, and at a constant value of β (= 1.0, 2.0, 3.0, 4.0), was investigated.

The results obtained from the study are as follows:Due to the random distribution of PMMA bone cement particles, a three-parameter Weibull distribution method was employed to analyze the discrete characteristic of the fracture property. For different values of β, when the characteristic crack was $\alpha_{\mathrm{ch}}^{*}=8d_{\mathrm{av}}$, tensile strength $f_{t}$ and fracture toughness $K_{\mathrm{IC}}$ with the smallest coefficient of variation ( CV ) were obtained. The minimum CV values for $f_{t}$ and $K_{\mathrm{IC}}$ were 0.1485 and 0.1492, respectively.The mean value μ (= 43.23 MPa), minimum value $f_{t}^{\min}$ (= 26.29 MPa), standard deviation σ (= 6.42 MPa) of tensile strength $f_{t}$, and these values of fracture toughness $K_{\mathrm{IC}}$ (μ = 1.77 $\mathrm{MPa}\cdot m^{1/2}$, $K_{\mathrm{IC}}^{\min}$ = 1.02 $\mathrm{MPa}\cdot m^{1/2}$, σ = 0.2644 $\mathrm{MPa}\cdot m^{1/2}$) were determined simultaneously from the three-parameter Weibull distribution method by using the characteristic crack $\alpha_{\mathrm{ch}}^{*}=8d_{\mathrm{av}}$ and the fictitious crack growth $\Delta \alpha_{\mathrm{fic}}=1.0d_{\mathrm{av}}$. Furthermore, the lower safety control index of PMMA bone cement was obtained based on the statistical properties of the minimum value of $f_{t}$ and $K_{\mathrm{IC}}$.The whole prediction breaking curve with 95% reliability for PMMA bone cement was obtained. Additionally, based on the simplified prediction model, the prediction line between peak load $P_{\max}$ and equivalent area $A_{e}^{1}\left(A_{e}^{2}\right)$ was obtained with 95% reliability. Nearly all experimental data are located within the scope of a 95% confidence interval. All experimental data were gained above the lower limit for safe prediction value.


## Figures and Tables

**Figure 1 polymers-14-03589-f001:**
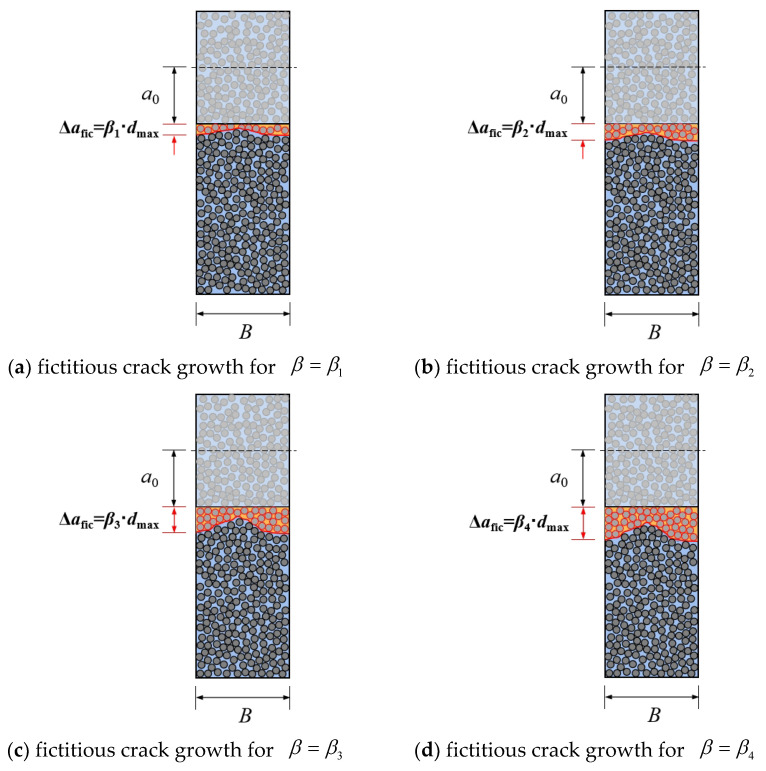
Discreteness of fictitious crack growth at peak loads.

**Figure 2 polymers-14-03589-f002:**
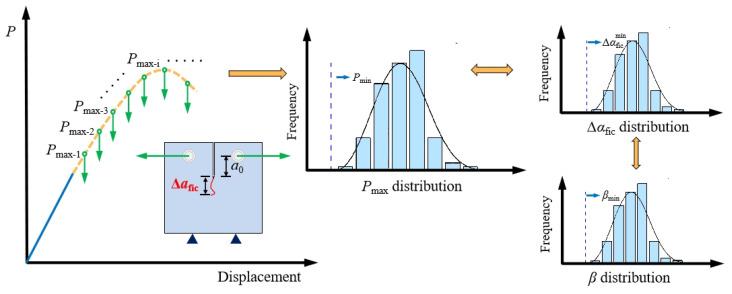
Distribution characteristics of the fracture strength parameters of PMMA bone cement.

**Figure 3 polymers-14-03589-f003:**
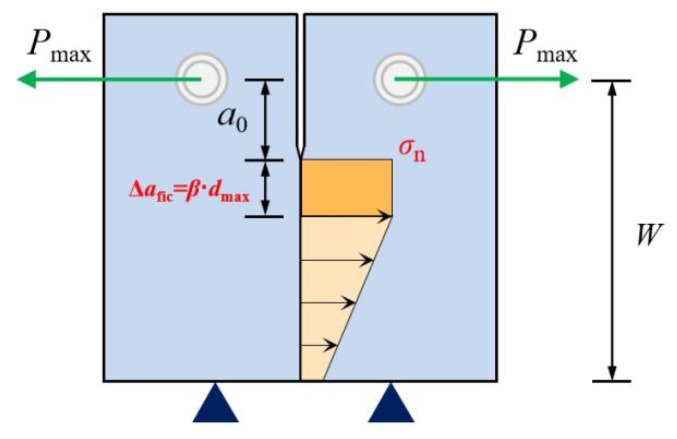
BEM analysis for PMMA bone cement.

**Figure 4 polymers-14-03589-f004:**
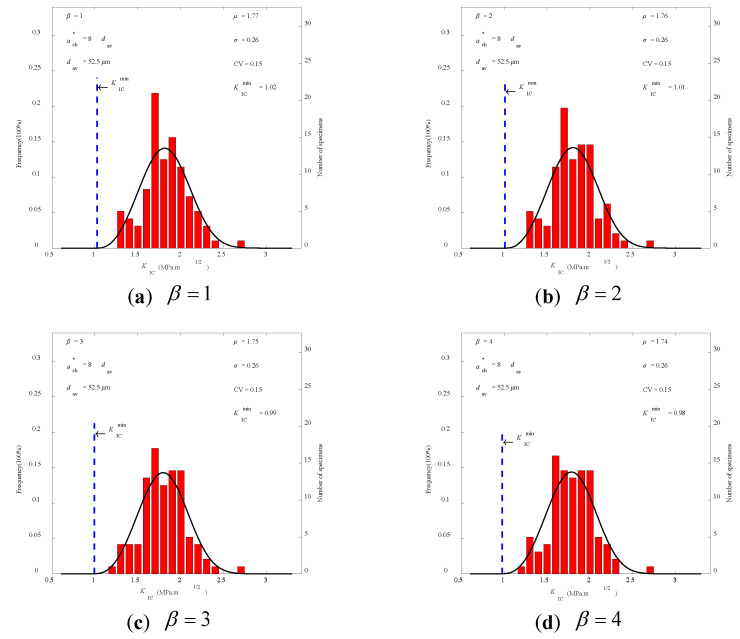
Three-parameter Weibull distribution of fracture toughness $K_{\mathrm{IC}}$ at $d_{\mathrm{av}}=52.5\;\mu m$.

**Figure 5 polymers-14-03589-f005:**
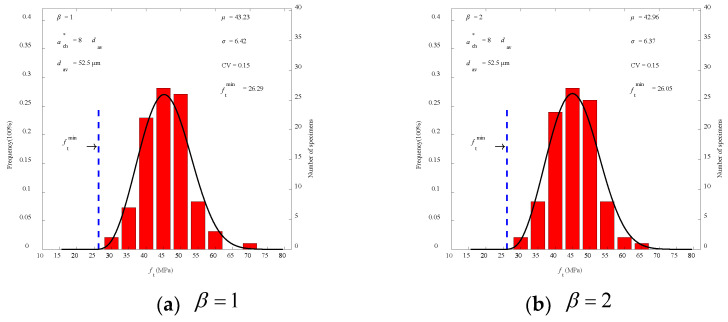
Three-parameter Weibull distribution of tensile strength $f_{t}$ at $d_{\mathrm{av}}=52.5\;\mu m$.

**Figure 6 polymers-14-03589-f006:**
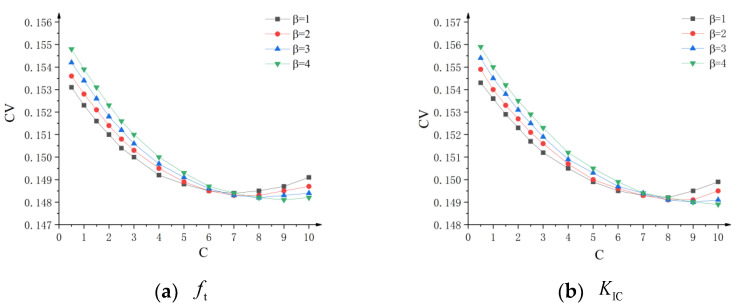
Variation curve of coefficient of variation CV for fracture toughness and tensile strength with *C* value at $d_{\mathrm{av}}=52.5\;\mu m$.

**Figure 7 polymers-14-03589-f007:**
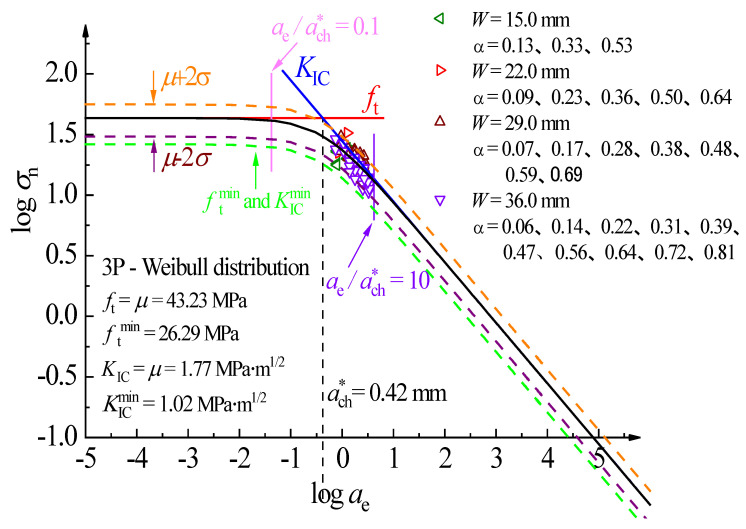
Destruction prediction full curve for $d_{\mathrm{av}}=52.5\;\mu m$, β=1, and C=8.

**Figure 8 polymers-14-03589-f008:**
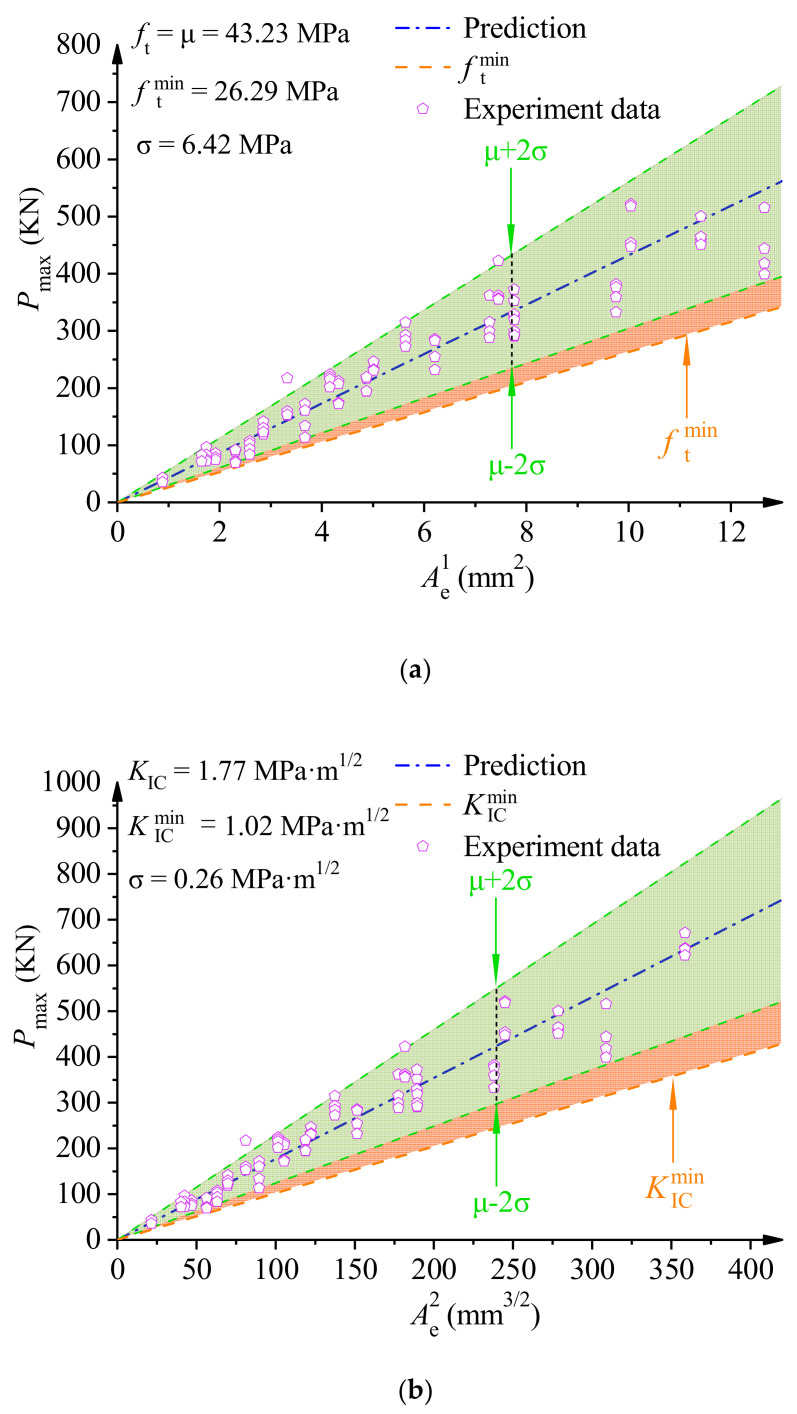
(**a**) Prediction curve of $f_{t}$ for $d_{\mathrm{av}}=52.5\;\mu m$, β=1, and C=8. (**b**) Prediction curve of $K_{\mathrm{IC}}$ for $d_{\mathrm{av}}=52.5\;\mu m$, β=1, and C=8.

**Figure 9 polymers-14-03589-f009:**
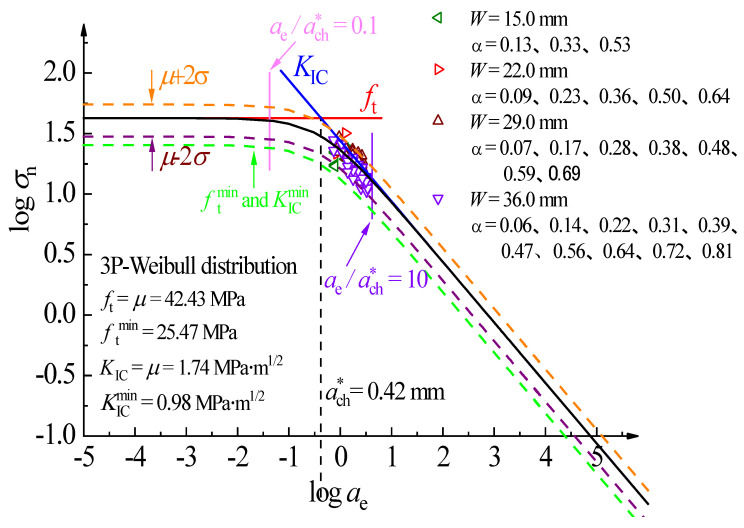
Destruction prediction full curve for $d_{\mathrm{av}}=52.5\;\mu m$, β=4, and C=8.

**Figure 10 polymers-14-03589-f010:**
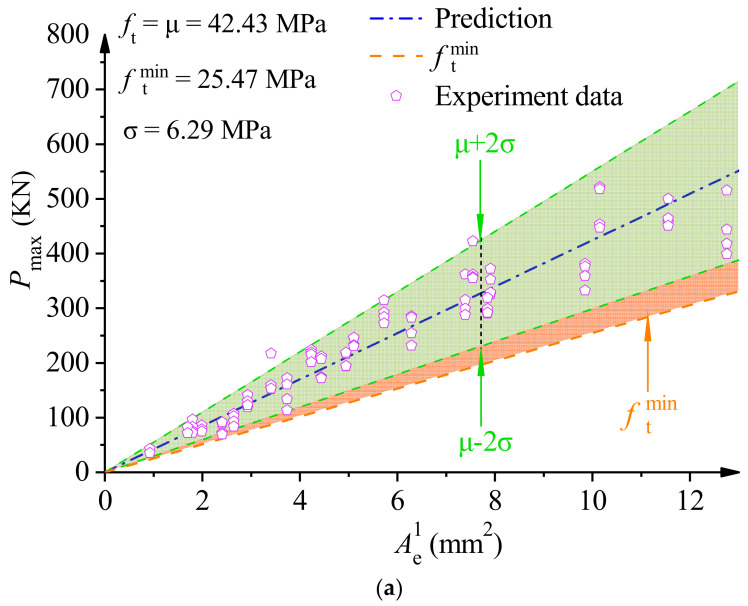
(**a**) Prediction curve of $f_{t}$ for $d_{\mathrm{av}}=52.5\;\mu m$, β=4, and C=8. (**b**) Prediction curve of $K_{\mathrm{IC}}$ for $d_{\mathrm{av}}=52.5\;\mu m$, β=4, and C=8.

**Table 1 polymers-14-03589-t001:** PMMA bone cement research table.

No.	Main Ingredient	Type of Test Specimen	Tensile Strength (MPa)	Fracture Toughness $\left(\mathrm{MPa}\cdot m^{1/2}\right)$	Source	Remark
1	Palacos^®^R	3-p-b *		2.70 ± 0.22	Kim [10]	
2	Simplex^®^P	WS *		2.15 ± 0.11	Biggs [11]	
3	Kyphon Xpede Bone Cement	WS *		1.11 ± 0.03	Pba [12]	
4	Palacos^®^R	WS *		1.85 ± 0.12	Lewis [13]	
5	Palacos^®^R CMW3 Osteopal	CNSR *		1.75 ± 0.07 1.92 ± 0.10	Lewis [14]	
6	Palacos^®^R	CNSR *		1.81 ± 0.14	Lewis [15]	
7	Simplex^®^P	CNSR *		1.30 ± 0.26^①^ 1.00 ± 0.05^②^ 1.01 ± 0.04^③^	Wang [16]	① UHM specimens ② CHM specimens ③ CFG specimens
8	Palacos^®^R CMW-1 Simplex P Zimmer D	CNSR *	33 ± 2 40 ± 8	1.59 ± 0.07^④^ 1.73 ± 0.17^⑤^	Kindt- Larsen [17]	④ Open bowl mixing ⑤ Vacuum mixing
9	Palacos^®^R	DB *	51.4 ± 3.47		Harper [18]	
10	IDX Palacos^®^R	4-p-b *	43.4 ± 1.6		Jellson [19]	
11	commercially available PMMA	DB *	44.7 ± 4.3		Harper [20]	
12	Palacos^®^R IDX IHX	DB *	42		Kjellson [21]	

Note *: 3-p-b: three-point bending specimens; WS: wedge-splitting specimens; CNSR: chevron notched short-rod specimens; DB: dumb-bell specimens; 4-p-b: four-point bending specimens; UHM: uncontrolled hand-mixed; CHM: controlled hand-mixed; CFG: centrifuged.

**Table 2 polymers-14-03589-t002:** Experimental results of WS samples.

Height (*W*) (mm)	Initial Crack Length $\left(a_{0})\;(\mathrm{mm}\right)$	Peak Load $\left(P_{\max})\;(N\right)$	Height (*W*) (mm)	Initial Crack Length $\left(a_{0})\;(\mathrm{mm}\right)$	$\mathrm{Peak}\;\mathrm{Load}\;(P_{\max})\;(N)$	Height (*W*) (mm)	Initial Crack Length $\left(a_{0})\;(\mathrm{mm}\right)$	$\mathrm{Peak}\;\mathrm{Load}\;(P_{\max})$ (N)
15	2	352.47	29	5	521.23	36	11	318.81
		325.47			518.00			298.50
		328.45			453.23			290.32
		372.29			447.03			291.82
	5	211.80		8	422.57		14	285.38
		207.34			361.23			282.65
		173.90			356.99			254.17
		171.68			354.99			231.63
	8	89.19		11	314.48		17	216.53
		91.43			292.38			218.52
		71.36			283.60			194.00
		69.13			272.27			194.49
22	2	499.78		14	223.65		20	171.69
		457.27			219.08			160.56
		463.97			215.19			133.80
		450.63			201.33			113.25
	5	361.51		17	140.92		23	107.06
		314.95			129.47			102.60
		299.04			118.85			92.95
		287.90			122.84			83.53
	8	245.13	36	20	96.19		26	82.54
		245.73			83.06			
		232.10			75.20			71.40
		230.31			72.18			
	11	216.92		2	693.65		29	42.91
		158.99			691.56			
		152.10			580.10			35.23
		152.80			528.34			
	14	84.77		5	515.45	\	\	\
		85.47			443.64			
		78.76			418.11			
		74.65			398.80			
29	2	670.70		8	381.21	\	\	\
		637.41			376.26			
		635.20			358.93			
		621.61			332.43			

**Table 3 polymers-14-03589-t003:** PMMA three-parameter Weibull distribution calculation results at $d_{\mathrm{av}}=52.5\;\mu m$.

$C=\alpha_{\mathrm{ch}}^{*}/d_{\mathrm{av}}$	$\beta =\Delta a_{\mathrm{fic}}/d_{\mathrm{av}}$	1	2	3	4
0.5	$f_{t}$	155.61	154.64	153.69	152.76
	$f_{t}^{\min}$	85.40	83.88	82.30	80.77
	σ	23.82	23.76	23.70	23.65
	CV	0.1531	0.1536	0.1542	0.1548
	$K_{\mathrm{IC}}$	1.59	1.58	1.57	1.57
	$K_{\mathrm{IC}}^{\min}$	0.86	0.79	0.78	0.77
	σ	0.2461	0.2454	0.2447	0.2441
	CV	0.1543	0.1549	0.1554	0.1559
1.0	$f_{t}$	110.90	110.21	109.53	108.87
	$f_{t}^{\min}$	61.59	60.56	59.43	58.39
	σ	16.89	16.84	16.80	16.76
	CV	0.1523	0.1528	0.1534	0.1539
	$K_{\mathrm{IC}}$	1.61	1.60	1.59	1.58
	$K_{\mathrm{IC}}^{\min}$	0.82	0.81	0.80	0.78
	σ	0.2468	0.2460	0.2453	0.2446
	CV	0.1536	0.1540	0.1545	0.1550
1.5	$f_{t}$	91.25	90.68	90.12	89.57
	$f_{t}^{\min}$	51.32	50.41	49.51	48.65
	σ	13.83	13.79	13.75	13.71
	CV	0.1516	0.1521	0.1526	0.1531
	$K_{\mathrm{IC}}$	1.62	1.61	1.60	1.59
	$K_{\mathrm{IC}}^{\min}$	0.84	0.82	0.81	0.80
	σ	0.2476	0.2467	0.2460	0.2452
	CV	0.1529	0.1533	0.1538	0.1542
2.0	$f_{t}$	79.62	79.13	78.64	78.16
	$f_{t}^{\min}$	45.26	44.50	43.71	42.95
	σ	12.02	11.98	11.94	11.91
	CV	0.1510	0.1514	0.1518	0.1523
	$K_{\mathrm{IC}}$	1.63	1.62	1.61	1.60
	$K_{\mathrm{IC}}^{\min}$	0.85	0.84	0.83	0.81
	σ	0.2485	0.2476	0.2467	0.2459
	CV	0.1523	0.1527	0.1531	0.1537
2.5	$f_{t}$	71.75	71.30	70.86	70.43
	$f_{t}^{\min}$	41.17	40.48	39.81	39.16
	σ	10.79	10.75	10.72	10.68
	CV	0.1504	0.1508	0.1512	0.1516
	$K_{\mathrm{IC}}$	1.64	1.63	1.62	1.61
	$K_{\mathrm{IC}}^{\min}$	0.87	0.85	0.84	0.83
	σ	0.2494	0.2485	0.2476	0.2467
	CV	0.1517	0.1521	0.1525	0.1532
3.0	$f_{t}$	65.98	65.57	65.16	64.77
	$f_{t}^{\min}$	38.18	37.58	37.00	36.39
	σ	9.90	9.85	9.82	9.78
	CV	0.1500	0.1503	0.1506	0.1510
	$K_{\mathrm{IC}}$	1.66	1.65	1.64	1.63
	$K_{\mathrm{IC}}^{\min}$	0.88	0.87	0.86	0.84
	σ	0.2505	0.2495	0.2485	0.2476
	CV	0.1512	0.1516	0.1519	0.1519
4.0	$f_{t}$	57.97	57.60	57.25	56.90
	$f_{t}^{\min}$	34.06	33.56	33.07	32.53
	σ	8.65	8.61	8.57	8.54
	CV	0.1492	0.1495	0.1497	0.1500
	$K_{\mathrm{IC}}$	1.68	1.67	1.66	1.65
	$K_{\mathrm{IC}}^{\min}$	0.91	0.90	0.88	0.87
	σ	0.2528	0.2516	0.2505	0.2495
	CV	0.1505	0.1507	0.1509	0.1512
5.0	$f_{t}$	52.57	52.24	51.92	51.60
	$f_{t}^{\min}$	31.28	30.85	30.41	29.97
	σ	7.82	7.78	7.74	7.70
	CV	0.1488	0.1489	0.1491	0.1493
	$K_{\mathrm{IC}}$	1.70	1.69	1.68	1.67
	$K_{\mathrm{IC}}^{\min}$	0.94	0.93	0.91	0.90
	σ	0.2554	0.2540	0.2528	0.2517
	CV	0.1499	0.1500	0.1503	0.1505
6.0	$f_{t}$	48.65	48.34	48.04	47.74
	$f_{t}^{\min}$	29.23	28.87	28.48	28.10
	σ	7.22	7.18	7.14	7.10
	CV	0.1485	0.1485	0.1486	0.1487
	$K_{\mathrm{IC}}$	1.73	1.72	1.71	1.69
	$K_{\mathrm{IC}}^{\min}$	0.97	0.96	0.94	0.93
	σ	0.2582	0.2567	0.2553	0.2540
	CV	0.1495	0.1496	0.1497	0.1499
7.0	$f_{t}$	45.63	45.34	45.06	44.78
	$f_{t}^{\min}$	27.62	27.30	27.00	26.64
	σ	6.77	6.73	6.68	6.64
	CV	0.1484	0.1483	0.1483	0.1484
	$K_{\mathrm{IC}}$	1.75	1.74	1.73	1.72
	$K_{\mathrm{IC}}^{\min}$	1.00	0.98	0.97	0.95
	σ	0.2612	0.2596	0.2581	0.2566
	CV	0.1493	0.1493	0.1494	0.1494
8.0	$f_{t}$	43.23	42.96	42.69	42.43
	$f_{t}^{\min}$	26.29	26.05	25.75	25.47
	σ	6.42	6.37	6.33	6.29
	CV	0.1485	0.1483	0.1482	0.1482
	$K_{\mathrm{IC}}$	1.77	1.76	1.75	1.74
	$K_{\mathrm{IC}}^{\min}$	1.02	1.01	0.99	0.98
	σ	0.2644	0.2626	0.2610	0.2594
	CV	0.1492	0.1491	0.1491	0.1492
9.0	$f_{t}$	41.27	41.01	40.75	40.50
	$f_{t}^{\min}$	25.18	24.95	24.71	24.47
	σ	6.14	6.09	6.04	6.00
	CV	0.1487	0.1485	0.1483	0.1481
	$K_{\mathrm{IC}}$	1.79	1.78	1.77	1.76
	$K_{\mathrm{IC}}^{\min}$	1.03	1.03	1.02	1.00
	σ	0.2683	0.2659	0.2640	0.2623
	CV	0.1495	0.1491	0.1490	0.1490
10.0	$f_{t}$	39.63	39.38	39.13	38.89
	$f_{t}^{\min}$	24.21	24.02	23.83	23.61
	σ	5.91	5.86	5.81	5.76
	CV	0.1491	0.1487	0.1484	0.1482
	$K_{\mathrm{IC}}$	1.82	1.80	1.79	1.78
	$K_{\mathrm{IC}}^{\min}$	1.05	1.04	1.04	1.03
	σ	0.2723	0.2698	0.2674	0.2654
	CV	0.1499	0.1495	0.1491	0.1489

## Data Availability

The data presented in this study are available on request from the corresponding author.

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
