# Peer review of "A Three-Parameter Weibull Distribution Method to Determine the Fracture Property of PMMA Bone Cement"

_polymers, 2022, doi:10.3390/polym14173589_

Round 1

Reviewer 1 Report

The paper was well conceived and prepared, however, it is not suitable in its present form.

The authors need to;

<< Report quantitative values of your findings in the abstract

<< How the authors need to use equation editors for the equations and make the fonts uniform throughout the text

<< Figures are better label accordingly for fig 1 (as a,b,c,d) where appropriate. The same observation for Fig. 4 and 5 respectively.

<< References are better placed at the end of the statement "[ ]

<< How authors need to cite at least three papers in polymers with similar work

Author Response

Dear Wei Zhang, Sixun Zheng

Editor in Chief

Polymers

On behalf of my co-authors, we thank you very much forgiving us an opportunity to revise our manuscript, we appreciated it or and reviewers very much for their positive and constructive comments and suggestions on our manuscript entitled "A three-parameter Weibull distribution method to determine the fracture property of PMMA bone cement”. (Manuscript ID: polymers-1833068).

We have studied reviewer’s comments carefully and have made revision which marked in red in the paper. We have tried our best to revise our manuscript according to the comments. Attached please find the revised version, which we would like to submit for your kind consideration.

We would like to express our great appreciation to you and reviewers for comments on our paper. Looking forward to hearing from you.

Yours sincerely,

Junfeng Guan

School of Civil Engineering and Communication, North China University of Water Resources and Electric Power

Reviewer 2 Report

No discussion and conclusions should be developed more. 

Author Response

(The authors gave the same response as above.)
